# Recycling of Selective Laser Sintering Waste Nylon Powders into Fused Filament Fabrication Parts Reinforced with Mg Particles

**DOI:** 10.3390/polym13132046

**Published:** 2021-06-22

**Authors:** Mohammad Uddin, Daniel Williams, Anton Blencowe

**Affiliations:** 1UniSA STEM, University of South Australia, Mawson Lakes, SA 5095, Australia; daniel.williams@unisa.edu.au; 2UniSA Health and Clinical Sciences, University of South Australia, Adelaide, SA 5000, Australia; anton.blencowe@unisa.edu.au

**Keywords:** SLS nylon waste, recycling, fused filament fabrication, Mg particles, 3D printing

## Abstract

This paper presents recycling of selective laser sintering (SLS) waste nylon into printable filaments and parts reinforced with Mg particles. Waste nylon and waste–Mg powder mixture with 2%, 4%, and 8% Mg to nylon were extruded into the filaments. Moisture absorption, differential scanning calorimetry, and melt flow index experiments were conducted to determine the thermal characteristics, while tensile and flexural tests were conducted to evaluate mechanical properties and failure mechanisms. The results were compared with off-the-self (OTS) nylon. Waste powder was found to be extrudable and printable as FFF filament. Waste filament diameter closely matched standard filament size, while exhibiting reduced moisture absorption. High melting and crystallisation temperature for the waste nylon demonstrated a degradation of the plastic during the SLS process. Young’s modulus and ultimate tensile strength for the waste filament increased by 1.6-fold compared to that for OTS, while Mg-composite filament surpassed the waste and OTS. Waste and Mg composite dog bone results showed an increase in strength and stiffness, but the ductility deteriorated. Both flexural strength and modulus for the waste nylon increased by 13% and 26%, respectively, over OTS, and the addition of Mg enhanced flexural strength by up to 5-fold at 8% Mg over the waste. Printed surface topography demonstrated that the waste and Mg composite filaments can print the parts with desired geometric shapes and acceptable surface texture. The findings showed that recycling waste SLS powder into FFF prints would be a viable and useful alternative to disposal, given its abundance.

## 1. Introduction

As an alternative to injection moulding, selective laser sintering (SLS) is a form of additive manufacturing (AM), where laser irradiation is employed to fuse and bind powdered plastic to form three-dimensional (3D) objects [1]. After the completion of an SLS print, a cube of loose powder is formed with the solidified printed components immersed inside. The models are retrieved by removing the loose powder, which is vacuumed back into the system. The unused powder accounts for 80–90% of the build volume and can be used for future prints [2]. However, the unused powder is physically and chemically altered by the laser’s heat, which causes deterioration in its mechanical properties and model appearance [3]. The most notable effects are high shrinkage of parts and a rough surface finish known as the ‘orange peel’ texture. However, this waste powder currently has no widespread means of recycling, and hence, eventually is disposed into landfill.

Researchers and engineers in industry have diagnosed the issue of aged SLS powder and suggested methods to reduce the aging effect, mainly through the addition of virgin powder. The recommended ratio of virgin powder to used powder is 20–50%, depending on the system [4,5]. This process can be repeated 2–3 times, and after that the powders become completely unusable and must be disposed [2]. An alternative to disposing of the waste SLS powders is to recycle them for use as the filaments for fused filament fabrication (FFF) prints. The FFF process is less sensitive to plastic quality, so printing with the waste powder filament would be a viable option that would offer a significant cost recovery. Generally, an average SLS build volume is 20 kg, and 25% of that powder is waste. Hence, a 5 kg of powder is wasted for each build volume [6]. This powder costs up to $180 per kg, so $900 of powder is wasted per build volume. FFF nylon filaments cost on average $100 per kilogram, meaning a reimbursement of $500 per build volume.

In the past, researchers investigated the challenge of fabricating recycled FFF filament from waste SLS powder [6,7,8]. Waste filament was reported to lower the melt flow rate, tensile strength, and Young’s modulus, which were attributed to the aging effect and repeated thermal cycles of waste powders during SLS [9].

However, a few studies have investigated the long-term printability of the filament, especially for substantial-sized models or as a replacement of regular filaments. Research regarding the inclusion of additives into FFF filament is widespread, with popular additives including carbon fibre [10], graphene [11], hemp [12], and tungsten carbide [6] to improve further mechanical properties. Recently, Antoniac et al. [13] studied magnesium (Mg)-reinforced PLA filament and demonstrated the printability and improvement in mechanical properties. Due to their superior mechanical and biological properties, it is possible that Mg particles could thus be added into SLS waste nylon to print different lightweight surgical instruments and accessories of complex geometric shapes with required mechanical strength and stiffness. SLS nylon powder complies with ISO 10993-5 and ISO 10993-10 associated with in vitro cytotoxicity and irritation, which also make nylon printed parts suitable for medical devices, such as casts and orthotics. In addition, Mg composite filament aims to show the potential for 3D-printed orthopedic devices, such as bone implants, cardiovascular stents, and scaffolds. However, little research has focused on the process viability of recycling of waste SLS nylon powders into usable FFF filaments reinforced with Mg particles for its practical applications.

Thus, the aim of this study was to investigate the entire process of creating FFF filaments from waste SLS powder, including powder analysis, extrusion, physical and thermal characteristics, mechanical testing, and printing. Waste SLS nylon powders were then reinforced with Mg powders to improve further mechanical properties of the filaments. Change in mechanical properties and surface integrity for the waste nylon–Mg composite filaments and FFF prints were compared with those for the waste and off-the-shelf (OTS) counterparts. As a case study, a jaw implant model was printed with the waste–Mg composite filament to demonstrate its suitability in biomedical application.

## 2. Materials and Methods

### 2.1. Material

The nylon 12 (PA12) powder (DuraForm ProX PA (SLS), 3DSystems, Rock Hill, SC, USA) used in this study was sourced from a local SLS printing company (Integrated Print Solutions (IPS 3), Adelaide, Australia). The plastic is a fine white powder of approximately 30 µm in diameter, with a stated melting temperature of 175–190 °C. The sintered part density is 0.95 g/cm^3^ with an ultimate tensile strength of 50 MPa, a tensile modulus of 1770 MPa, elongation at break of 22%, a flexural strength of 60 MPa, and flexural modulus of 1650 MPa. A commercially available off-the-shelf nylon filament was tested as a comparison, showing the properties of a traditional FFF filament. The OTS nylon filament (ePA) was sourced from eSUN Pty Ltd., Shenzhen, China (the grade of plastic and additives were unspecified). Mg powder was chosen as the reinforcing filler material for the composite filaments. The Mg was >99.9% pure and 200 mesh in size (supplied by Aus Chem Source, Beckenham, Australia). The density of the Mg powder was 1.738 g/cm^3^ and the melting temperature 648.8 °C. Three different ratios of Mg (2%, 4%, and 8% in weight) to waste nylon powder were considered. Mg powders of appropriate quantity were mixed with the waste nylon powder in a container and shaken vigorously to disperse Mg evenly amongst the nylon powder. For each sample, 250 g of the nylon–Mg powder mixture was created, and such quantity was enough to create the filaments required for characterisation and mechanical tests. Before fabricating the filaments, the powder mixtures were all dehydrated for 6 h to remove any moisture.

### 2.2. Filament Fabrication

A filament extrusion system (Filastruder 2020, Snellville, GA, USA) was used to create the filament from the nylon and magnesium powder mixtures, following ASTM 52903-1-20 standard. The Filastruder was a single-screw filament extruder and was chosen due to its low cost and built-in nozzle mesh filter preventing any large particles from contaminating the filament. The distance between the extruder and spooler was 600 mm horizontally and 700 mm vertically between the extruder and the spooler sensor. The extrusion temperature varied between 205 °C and 220 °C for the pure nylon and composite filaments. In this study, we could not extrude filaments from virgin SLS powders because of their high melt flow rate (i.e., very low viscosity). Therefore, the relevant mechanical properties of virgin SLS powder were cited from the manufacturer’s data sheet and compared with waste nylon and composite.

### 2.3. Thermal Characterisation

The melt flow index (MFI) for pure nylon and composites was measured by using a custom-made MFI machine following the ASTM D-1238 standard. The barrel temperature was 235 °C with a 2 min plastic preheating followed by a weight of 2.16 kg being applied on the piston. A differential scanning calorimetry (DSC) machine (Netzsch STA 449 Jupiter, Netzsch-Gerätebau GmbH, Selb, Germany) was used to determine the transition temperatures including crystallisation and melting temperatures of the samples. The DSC testing parameters were a temperature range of 30 °C to 230 °C, with a heating rate of 10 °C/min, using a graphite crucible under argon. Two heating and cooling cycles were run to eliminate the thermal history of the plastics.

### 2.4. Part Fabrication

The extruded filament was printed using a 3D printer (Flashforge Creator Pro, FlashForge Corporation, Jinhua, Zhejiang, China) to create parts for mechanical testing along with determining print surface characteristics and visual inspection. The extruder had been an upgraded Flexion high-temperature extruder capable of temperatures above 250 °C and adjustable tension on the filament. The print settings were: extruder temperature = 243 °C, bed temperature = 85 °C, print speed = 50 mm/s, layer height = 0.2 mm, and raster angle = 45° for all models. The infill percentage used for the tensile and flexural specimens was 100%, aiming to recreate a solid plastic section. Samples were fabricated on the XY plane of the print bed where the build was along the Z direction. The benchmark models, however, utilized 20% infill as a more realistic percentage for standard models. The print bed was the default Flashforge platform surface with a Magigoo PA (Tariq-al-Kasim, Malta) glue stick also used to adhere the nylon to the bed as it is often prone to warping due to the thermal shrinkage.

### 2.5. Mechanical Testing

Tensile and three-point flexural experiments were conducted using an Instron 5567 universal testing machine (Instron, Norwood, MA, USA). Tensile tests were conducted for both filament and printed dog bone specimens by following the ASTM D638 standard. A short section of filament with a gauge length of 50 mm was clamped between the grippers and tested until necking and stretching of the filament. As the grippers crushed the filament, a cloth was added to distribute the clamping force between the gripper and the filament. The crosshead speed was 150 mm/min and the test ended at an extension of 200 mm. This high crosshead speed was chosen as preliminary experimentations found a minimal difference between the tensile properties at different crosshead speeds. Dog bone tensile testing used the 3D-printed samples. The crosshead speed was 10 mm/min and the test ended when the specimen failed. Mechanical properties such as yield strength, Young’s Modulus, ultimate tensile strength, and elongation at break obtained from test data were analysed. Flexural experiments were performed on the printed samples of 80 × 10 × 4 mm^3^ by following the ASTM D790 standard. The crosshead speed was 10 mm/min and the test ended at a displacement of 15 mm. The key results to be investigated were flexural strength and flexural modulus.

### 2.6. Powder and Surface Characterisation

Moisture absorption and diameter of the filaments from waste and waste–Mg composite powders were measured and assessed. Surface topographies of the filaments and failure zones of the print samples after mechanical testing were characterised via scanning electron microscopy (SEM) (Merlin, Carl Zeiss Co., Jena, Germany). Prior to SEM, the samples were platinum coated to obtain detailed characteristic features. Surface roughness (R_a_ and R_t_) of the prints was measured by using a 3D laser scanning confocal microscope (Olympus’s LEXT OLS5100, Tokyo, Japan).

## 3. Results

### 3.1. Characterisations of Nylon and Mg Powders

Figure 1 shows the waste powder and waste–Mg composite powder mixtures. The increase in Mg content darkens the colour of the mixtures. The Mg powder sourced had a mesh size of 200, making the particles a maximum of 0.074 mm in diameter, which was lower than the FFF 3D printer nozzle size of 0.4 mm. If any particle is larger than the nozzle, blockages are possible, making the filament unusable. SEM images of the nylon, Mg powder, and 8% Mg powder mixture were taken to understand the distribution and size of the particles (Figure 2). The waste nylon powder particles varied in size (from 13 to 118 µm) and seemed spherical in shape (Figure 2a), while Mg particles were flake-like with sharper edges and varied between 18–180 µm (Figure 2b). Figure 2c shows an SEM image of 8% Mg mixture, exhibiting the distribution of nylon and Mg powders. Figure 2d further shows the uneven distribution of waste nylon powder. This may be due to the fact that during the SLS process the powders underwent manual handling, repeated sintering, and thermal cycles (i.e., heating and cooling), which may have increased molecular weights and thermal properties. This has been observed by other researchers [9].

### 3.2. Filament Physical Characteristics

The extruder had difficulty extruding the nylon powder, as it was optimised for traditional plastic pellets. Adding small amounts of powder proved an effective method, as larger amounts would bridge and arch in the hopper due to its shape and friction between the powder particles. This phenomenon was observed by [14], and powder agitation to ensure correct flow during extrusion was suggested. The extrusion temperature had the greatest impact on the filament diameter, and after initial trials, an extrusion temperate of 220 °C enabled an optimal diameter for the waste filament (Figure 3a). On the other hand, the extrusion temperature for the nylon–Mg filament was lower compared to that for the waste filament. For 2% and 4% Mg filaments it was 206.5 °C and was lowered to 204 °C for 8% Mg filaments. This may be because Mg powders absorb heat internally and the heat capacity increased with the increase of Mg powder content. Mg powders were found to be visible within the filaments, giving the filament a light grey colour and an increase in darkness (Figure 3b). Surface texture of the filaments was observed under SEM, showing a substantial texture change between the 2% and 8% Mg filaments. As presented in Figure 3c–e, 8% Mg composite filament exhibited a very rough surface texture due to a higher percentage of Mg powders in the nylon matrix (Figure 3e). Mg powders were found to agglomerate, protrude, and overlap with each other due to the nonuniformity in size and thermal property of the nylon and Mg powders being extruded through the nozzle.

The extruded filament diameter was measured over a meter length to determine its tolerance variation from the recommended diameter of 1.75 ± 0.05 mm [15]. As summarized in Table 1, the diameter of the waste nylon filaments (1.74 ± 0.05 mm) was comparable to that of OTS (1.74 ± 0.02 mm). However, the diameter of the composite filaments was found to slightly decrease as the Mg powders increased from 2% Mg (1.74 ± 0.03 mm) to 4% Mg (1.7 ± 0.02 mm) and 8% (1.71 ± 0.01 mm). Moreover, the variation of diameter for the composite filaments was less than 2% and fell within the recommend diameter tolerance. Therefore, this result again implies that the waste and composite filaments can be fabricated with suitable geometric dimensions for further use in FFF printing.

The moisture absorption capacity of the powder and filaments were determined by dehydrating the samples followed by exposing them to the atmosphere and recording their periodical weight change. As can be seen in Table 1, the virgin powder, waste powder, and waste filament all absorbed almost the same moisture of 0.4% over 48 h, with the majority occurring within the first 12 h. This was quite lower than that for the OTS filament which absorbed about 0.95% moisture over 48 h. This means that filaments made of waste nylons and their Mg composites can be stored in self for a longer period without potential degradations due to moisture absorption.

### 3.3. Melt Flow Index

The melt flow index was measured to determine the plastic flow rates, finding the amount of plastic extruded over a 10 min period. As is summarized in Table 2, the virgin nylon powder had the highest flow rate of 52.01 g/10 min, while the waste powder was lower (20.38 g/10 min), suggesting the degradation of the nylon through thermal cycles during the SLS process. When the waste powder was formed into filament, the flowrate increased again. This is expected to be beneficial for the FFF process as Mg particles within the filament will not be clogged in the extrusion nozzle due to the increase of Mg particle concentration. The high MFI for the virgin powder explains the difficulty in extrusion and making filaments, as the viscosity is too low to be conveyed out the nozzle. Other researchers have conducted MFI testing on virgin and waste SLS powder using the same test conditions and found similar results. Pham et al. [2] found 51.5 g/10 min for virgin powder and Tiwari and Kumar [16] 33.1 g/10 min for waste powder. It is to be noted that neither of these studies used the same grade of powder, and so the SLS conditions of the waste powders were different.

As can be seen in Table 2, the increased Mg content reduced the flow rate because the Mg did not melt, meaning the plastic had to flow around it, drawing Mg particles out the die. This also caused the diameter of the filaments to reduce (see Table 1) as the plastic was heated for longer, and Mg particles retained more heat. Therefore, it should be mentioned that the appropriate percentage of Mg in the composite filament should not be decided based on the flowrate, but on the required balance strength and ductility of the final part. This may also imply that the printing parameters may need to be adjusted depending on the filament materials. Such requirements are expected to push the current boundary of FFF, which will enable a single printer to accommodate a wide range of parameters and materials to be printed. The OTS filament had showed the lowest MFI and the lowest standard deviation. This result is aligned with the MFI reported by the supplier of the OTS (eSun), which further proves that the custom-built MFI machine is reasonably accurate.

### 3.4. Differential Scanning Calorimetry (DSC)

Figure 4 shows DSC curves for powders and filaments, including the virgin, waste, and Mg composites. The shape of the DSC curves shows that the nylon plastic had a semi-crystalline structure, and the peaks represent the two critical temperatures. The crystallisation temperature (*T_c_*) is the peak of the cooling cycle and melting temperature (*T_m_*) the peak of the heating cycle. While waste nylons show single *T_m_* peaks, nylon–Mg composites exhibit double *T_m_* peaks in heating cycles. On the other hand, single *T_c_* peaks are observed for both waste nylons and their Mg composites in cooling cycles.

Table 3 summarizes DSC results for the virgin, waste and Mg composites. It can be seen from Table 3 and Figure 4 that *T_c_* for the waste powder (147.4 °C) and filament (153.25 °C) increased as compared to that for the virgin powder (143.31 °C). A similar trend for *T_m_* was observed across all the samples. Most importantly, the range between *T_c_* and *T_m_* for the waste filament (*T_m_*: 179.05 − *T_c_*: 153.25 = 25.8 °C) reduced significantly as compared to that for the waste (*T_m_*: 191.7 − *T_c_*: 147.4 = 44.3 °C) and virgin powder (*T_m_*: 178.07 − *T_c_*: 143.31 = 34.76 °C). These results demonstrate that the additional heating cycles affected the crystallisation–melting behaviour of the waste plastic. A smaller range is recommended to minimize the impact of shrinkage, making the waste nylon desirable for FFF printing [6].

The nylon–Mg composite filaments were found to exhibit almost similar melting (*T_m_* ~ 176 °C) and crystallisation (*T_c_* ~ 153 °C) temperatures and a corresponding temperature range (~26 °C) to the waste filament, suggesting that the addition of Mg powders into the nylon matrix has no effect on key thermal properties of waste nylon filaments. In other words, due to their smaller size and concentration, the Mg powders caused little or no restrictions in the movement and reordering of molecular chains of nylons under thermal loading.

As can be seen in Figure 4, in the heating cycles of all nylon–Mg composites, two melting peaks can be observed: the first melting peak occurs at 169 °C and the second peak at 176 °C. Note that the first heating cycle did not show any double melting peaks for nylon–Mg composites, therefore, DSC curves for the second heating and cooling cycle of the composites are presented in Figure 4. The double melting peaks for the second cycle could be related to the underlying recrystallisation–melting process. In other words, this could be due to the difference in the cooling rate causing the degree of crystallinity within the composite filament sample used in the first cycle and second cycle. The first smaller melting peak shown in Figure 4 can be attributed to the presence of relatively tiny crystals with thinner lamellae and already rearranged molecular chains caused by the first cycle; therefore, the plastic required a relatively lower temperature to initiate the melting. A similar phenomenon with the appearance of double melting peaks in the second heating cycle for carbon fibre-reinforced PA12 composite filaments were observed by other researchers [8].

### 3.5. Tensile Properties

The tensile tests were conducted on the filament and standard dog bone shape samples, which determined the mechanical properties. The filament samples did not break during all tests, so the elongation at break could not be ascertained and hence is not presented in this paper. Figure 5 shows representative stress–strain profiles obtained from filament tensile tests. Table 4 summarizes tensile properties extracted from stress–strain data for the filament and dog bone tests.

It can be seen from Table 4 that while the yield strength for the OTS and waste filament remained almost same, Young’s modulus and ultimate tensile strength (UTS) for the waste filament increased by about 1.6-fold (from 594.76 MPa to 938.3 MPa) and about 1.5-fold (from 30.96 MPa to 46.20 MPa), respectively, when compared with that of OTS. With the addition of Mg powders into waste nylon, yield strength, Young’s modulus, and UTS all appeared to increase and the improvement was found to be significant for 4% Mg filament. As the amount of Mg increased further from 4% up to 8%, however, such improvement seemed to decline by exhibiting a reduction of yield strength by 10% (from 32.23 MPa to 28.78 MPa) and Young’s modulus by 16% (from 1065 MPa to 892 MPa), and UTS by 22% (from 47.04 MPa to 36.69 MPa). This implies that higher percentages of Mg into the composite filament are likely to deteriorate mechanical properties.

On the other hand, the dog bone test results presented in Table 4 show that the yield strength, Young’s modulus, and UTS for the waste sample increased compared to the OTS sample. A similar trend was observed for the Mg composite samples. For instance, the yield strength, Young’s modulus, and UTS for the waste increased by 6%, 8% and 6%, respectively, when compared with that of OTS. They further increased by 38%, 22%, and 12% at with the 8% Mg composite sample. The results imply that the addition of Mg particles improved mechanical properties significantly.

As compared to the OTS sample (~106%), the waste sample showed significantly lower elongation at break (~21%). This may be because the molecular chains of waste nylon have been strengthened further due to thermal cycles in the SLS process. As the Mg increased, the elongation at break further dropped by ~10% in the case of the 8% Mg composite sample. This again indicates that Mg powders enhance mechanical strength but reduce the ductility.

As for the elongation at break, compared to the OTS and waste, the Mg addition made the plastic stiffer, causing it to break with less elongation. Figure 6 exhibits SEM imaging of failure areas of dog bone samples. While OTS and waste nylon layers had stretched significantly (Figure 6a,b), the Mg composite samples showed less stretching at the microscopic level (Figure 6c–e). In other words, Mg particles were bonded together within nylon, thus preventing further stretching or elongation of molecular chains of nylon until failure. It is, however, to be noted that a few Mg particles in places are found to separate from the nylon, indicating the poor particle interface with the nylon matrix (Figure 6d). This could be due to a thermal mismatch between Mg and nylon during the cooling of dog bone samples being printed from the composite filaments. This would be more likely to occur for the samples with higher percentage of Mg particles.

It is also seen from Table 4 that the yield strength and Young’s modulus obtained from the dog bone tests were found to be higher than that obtained from the filament tests, except the UTS for which the filament tests showed slightly higher values. Therefore, it is imperative to say that perhaps both filament as well as dog bone tensile samples must be tested to determine the overall mechanical properties (strength and ductility) of 3D printing material.

### 3.6. Flexural Properties

Figure 7 shows flexural stress–strain profiles obtained from three-point bending tests for different printed samples. The flexural test provided the most interesting results, with the waste and OTS exhibiting a similar trend of stress–strain curves, while the curves for Mg composites shifted upward. As summarized in Table 5, both the flexural strength and Young’s modulus for the waste nylon increased by 13% (from 30.91 to 34.81 MPa) and by 26% (from 202.59 to 255.8 MPa), respectively, when compared with that of OTS. As shown in Table 5, with the increase of Mg percentage, both the flexural strength and Young’s modulus for the composites increased. For example, the flexural strength for 2% Mg was 3 times greater than the waste, increasing to 3 times higher for the 8% Mg, reaching about 96 MPa. Similarly, the flexural modulus for 2% Mg was 4 times the waste, increasing to 6 times with 8% Mg, reaching about 1397 MPa. These results clearly show that even a small percentage of Mg addition into the nylon matrix drastically enhances the flexural strength and modulus, increasing the stiffness of the samples, making them more resistant to bending.

However, the flexural strength for the waste and OTS plastics were 50% lower than those obtained from the manufacturer (60 MPa for the waste and 57 MPa for OTS). A similar trend was observed for the flexural modulus, as compared to 1650 MPa for the virgin powder and 1495 MPa for the OTS obtained from the manufacturer [17]. The more likely reason for such a large difference in flexural properties was the 3D-printed nature of the flexural samples used in our study as compared to injection moulded samples used in the literature.

### 3.7. Analysis of Print Quality

Benchmark models consisting of different geometric surface (flat, curve) and shape (cuboid, spherical) characteristics were printed with the waste and composite filaments to evaluate the quality of their prints. As can be seen in Figure 8, the waste and Mg composite filaments were able to print good quality parts resembling the desired geometric shapes. While the OTS print showed a cleaner surface (Figure 8a), the waste print demonstrated signs of thin plastic stringing (i.e., protruded layers) at the corners or intersecting areas on the print surfaces (Figure 8b). Similar surface irregularities were noticed on Mg benchmark print surfaces as well (Figure 8c–e). Due to Mg particles, the models seemed to have a relatively smooth top surface and the print’s colour became darker with the increase in percentage of Mg.

The layer bonding varied between the samples. When magnified using SEM imaging, the print layers for the waste print were not bonded and aligned together properly and gaps and pores were noticed in places, resulting in high surface troughs (Figure 9). However, very regularly aligned and bonded layers were observed for the 4% Mg print. This result is highly consistent with mechanical property enhancement (tensile and flexural strength) for the waste + higher Mg percentage composites as presented in Table 4 and Table 5.

Figure 10 shows 3D surface topography of the samples printed with OTS, waste, and waste–Mg composite filaments. OTS and waste prints showed relatively smoother surface topography (Figure 10a,b). Mg concentration of as low as 2% caused an increase in surface texture (Figure 10c). However, when Mg concentration further increased from 2%, the waste–Mg print surface became smoother (Figure 10d). This was quite evident in the case of the 8% Mg print (Figure 10e). Surface roughness R_a_ and R_t_ are presented in Figure 11, and they follow the same trend and are consistent with surface topography. Higher Mg in the composite means less shrinkage and better bonding between the layers due to consistent thermal distribution, thus making the surface smoother. The results indicate that Mg particles can be added to the waste nylon to improve both surface quality and mechanical properties.

Despite the overwhelming surface quality being achieved, the prints may often still require further postprocessing to meet functional and aesthetic requirements [18]. To demonstrate this, a real-world jaw implant model was printed with 4% Mg filaments and postprocessed by cutting to remove support and unwanted artifacts. It can be seen from Figure 12 that surface irregularities and stringing can be successfully removed, leaving a very smoother surface finish on the final print product.

## 4. Discussion

Recycling of SLS waste nylon into FFF filaments and parts seems a plausible approach to recover revenue and protect the environment from potential contamination of landfills. In this study, we demonstrated that the waste nylon powders underwent thermal cycles that caused a degradation of properties. Repeating heating and cooling in SLS affected the arrangement of molecular chains, crystallinity, size, and shapes of powders [2,19]. This has been evidenced from nonuniform size distribution (Figure 2). Due to repeated sintering, the melting and crystallisation temperature of waste powder had increased as compared to that of virgin (unused) powder, and as a result, a higher extrusion temperature was often required for waste powder to extrude out the filament. Despite this, interestingly, the range between the melting and crystallisation temperature, as evidenced from DSC curves (Figure 4), was significantly lower, which provides a beneficial effect on the printability by minimising shrinkage and warpage. A similar trend was noticed for nylon–Mg composite filaments with the characteristic of double melting peaks in the second heating cycle. This finding was consistent with the MFI result (Table 2), where the waste filaments showed a lower melt flow rate than the virgin, with even a further decrease when Mg particles were added. This could be due to the combined thermal effect nylon powder underwent during SLS as well as the extrusion process. Filament size tolerance is a crucial parameter for the filament to be practically used by a standard FFF printer.

We demonstrated that the diameter of the waste filament (1.74 ± 0.05 mm) was comparable to the industry recommend size (1.75 ± 0.05) [20], except for slightly smaller filament sizes for nylon–Mg composite filaments due to more uniform shrinkage of filament layers. A relatively low moisture absorption capacity (of 0.4% over 48 h as opposed to OTS nylon or virgin) proved that the filaments can have a higher shelf life without deteriorating their chemical and physical characteristics. Surface treatment such as additional coating can be applied to nylon filaments to prevent moisture absorption but at the cost of a slight reduction in stiffness and strength [21]. Thus, SLS waste filaments would not incur additional expenditure for postprocessing.

Tensile filament and dog bone test results revealed an increase in yield strength, Young’s modulus, and UTS for the waste nylon and nylon–Mg composites as compared to OTS nylon, while the maximum improvement seemed to occur for the 4% Mg composite. As alluded to in Figure 6, the interaction between nylon and Mg powder can be interpreted as: initially, at a low percentage of Mg, the plastic provided a strong matrix for Mg, providing enough space between the particles to bond. As the percentage of Mg increases, the area of plastic bonding decreases, hence Mg cannot bond to itself, causing it to become weaker.

Slightly lower Young’s modulus and UTS values for the waste filament as compared to our results presented in Table 4 was reported in the literature. For example, Kumar and Czekanski [6] reported a UTS of 25 MPa (45% lower) and Young’s modulus of 700 MPa (20% lower). The reasons for these discrepancies are possibly due to the difference in the type of waste SLS powder used to make the filament and their thermal history and levels of deterioration. In the past, researchers tested injection molded dog bone specimens made from waste nylon pellets. For instance, Feng et al. [7] found a similar UTS to our dog bone results (of 37 MPa), while Wang et al. [8] reported a slightly higher UTS (of 43 MPa). Thus, the combination of filament and dog bone tensile tests was able to characterise the tensile properties of the nylon (Table 4). By comparing our results with the literature, the filament tests seemed to provide an accurate yield strength and Young’s modulus while the dog bone found an accurate elongation at break.

Flexural strength and modulus followed a similar trend. This was further supported by SEM imaging of failure mechanisms where the waste had lower stretching and Mg particles were highly bonded with nylon. The results clearly outlined that the waste nylon and its Mg composites could provide suitable mechanical integrity to the printed parts. Increase in stiffness and strength by reinforced WC particles into a nylon matrix was reported by other researchers [7]. It is to be noted that, in this study, we could not fabricate filaments and parts from virgin SLS nylon powders due to their very low viscosity, and a direct comparison of mechanical properties between virgin and waste nylon filaments was not presented. Instead, we compared the results with respect to OTS nylon (commercially available PA12 nylon), which may not exactly match with the grade of actual SLS waste used, and therefore, a caution has to be taken in interpreting the findings. Nevertheless, it is quite obvious that, due to degradation via thermal cycles, mechanical properties of SLS waste will deteriorate to some extent as compared to virgin SLS of the same grade or stock [7].

We demonstrated that the waste filament was able to produce the parts of different geometric shapes with acceptable accuracy and surface topography, except for surface stringing or protruded plastic debris left on the print, which can be removed via sanding. Due to the inherent nature of the layer-by-layer process, 3D-printed parts are often left with an irregular surface texture. However, the waste nylon–Mg composite prints were found to exhibit a very smooth surface texture, and this was prominent at a higher Mg concentration (8% Mg) because of more uniform layers and bonding. Thus, the printed part with the waste nylon together with reinforcing particles may require minimum postprocessing. Our study has shown that the recycled SLS nylon has a great potential for use in FFF prints and reinforcement by Mg can further enhance biomechanical integrity of the parts. Though we have demonstrated the process for nylon with Mg, other biomaterials such as carbon fibre and ceramic particles can be added to improve the properties. This will open up the new ways of fabricating low-strength load bearing medical devices (e.g., implants, orthotics) and healthcare instruments (e.g., surgical accessories—catheters, tubes) meeting both mechanical strength and biological requirements, thus saving costs and protecting environment [1].

Future research should focus on evaluating in vitro mechanical and biocompatibility properties of a real-world-printed medical part to demonstrate suitability and functional efficacy in medical applications. Optimisation of Mg concentration into nylon would be another aspect to be considered to determine an optimum mechanical and surface integrity of the print. Anisotropy in mechanical properties is a common phenomenon in 3D printing [22]. In this study, we printed and tested all tensile dog samples and parts at a constant single print setting, as outlined in Section 2. Variation in print direction, speed, and layer height can affect the print quality and properties as well. Developing a greater understanding on the printability of SLS waste nylon into FFF parts under a varied parameter setting would be an interesting subject of further study [23].

## 5. Conclusions

This study investigated the recycling of waste SLS nylon powder into its FFF printability. Waste nylon powder was found to be extrudable and printable as FFF filament, and the addition of Mg powder created a composite filament that showed enhanced properties from the pure waste nylon. Waste filament diameter closely matched standard filament size (1.75 ± 0.05 mm) while exhibiting reduced moisture absorption. High melting and crystallisation temperatures for the waste nylon and Mg composite filaments demonstrated a degradation of the plastic after multiple heating cycles during the SLS process. Tensile tests showed that mechanical properties of the waste nylon were comparable or even better than OTS (commercially available nylon filament). Young’s modulus and UTS values for the waste filament increased by 1.6-fold compared with that of OTS. Mg composite filament further improved the properties with a maximum enhancement occurring at 4% Mg content. Dog bone results showed an increase in strength and stiffness for the waste and Mg composite samples over OTS, but the ductility deteriorated by decreasing the elongation at break. Three-point bending test results revealed that both flexural strength and modulus for the waste nylon increased by 13% and by 26%, respectively, when comparing with that of OTS. The Mg composite samples further enhanced flexural properties by up to 5-fold at 8% Mg content over the waste filament. Surface topography analysis demonstrated that the waste and Mg composite filaments can print the parts with the desired geometric shapes and acceptable surface texture/finish with little postprocessing. Due to enhanced mechanical properties and printability, it is imperative to say that the prints made of waste SLS nylon combined with Mg particles can be potentially used in many applications, including as medical components in healthcare and hospitals. Further, salvaging SLS waste nylon into FFF printing could be a viable route to support a circular economy and sustainability in future additive manufacturing.

## Figures and Tables

**Figure 1 polymers-13-02046-f001:**
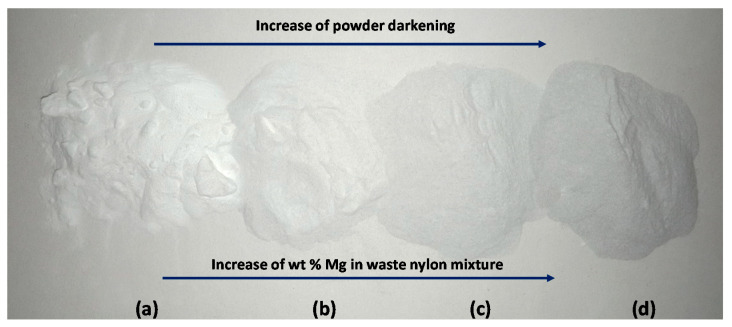
(**a**) Waste nylon; (**b**) 2% Mg; (**c**) 4% Mg; (**d**) 8% Mg powders in waste nylon mixtures.

**Figure 2 polymers-13-02046-f002:**
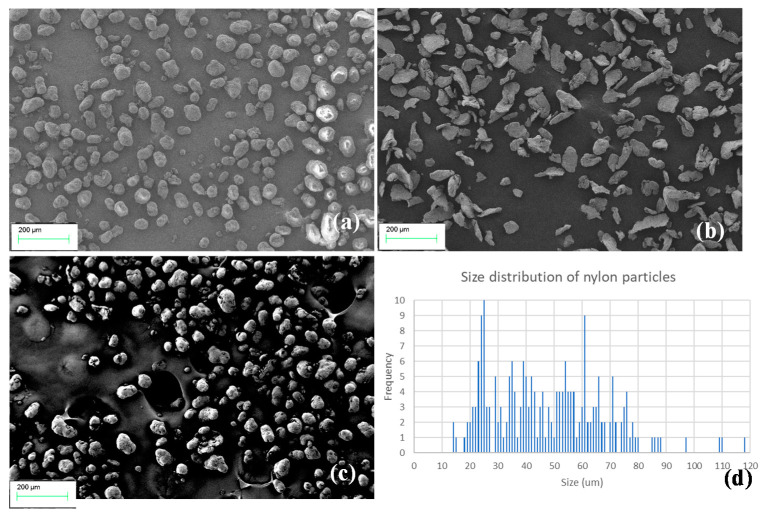
SEM images: (**a**) waste nylon powder; (**b**) Mg powder; (**c**) 8% Mg mixture; (**d**) size distribution of waste nylon powder.

**Figure 3 polymers-13-02046-f003:**
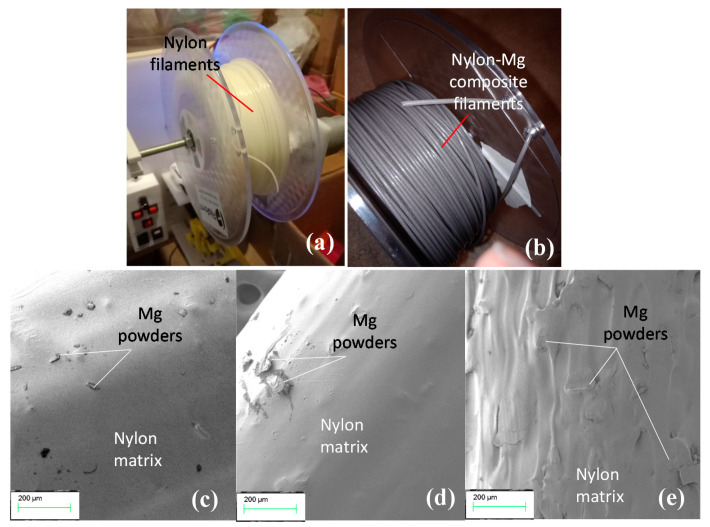
(**a**) Extruded waste nylon filaments; (**b**) Mg composite filaments spools; and SEM imaging of surface topography for (**c**) 2% Mg, (**d**) 4% Mg, and (**e**) 8% Mg composite filaments.

**Figure 4 polymers-13-02046-f004:**
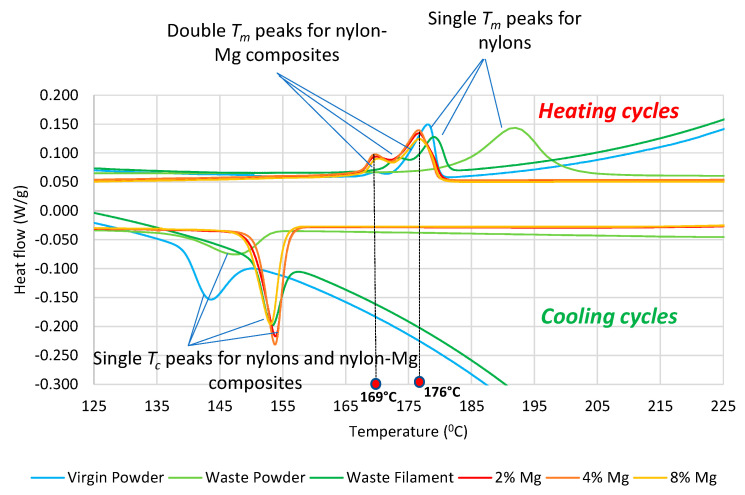
DSC curves for nylon and nylon–Mg composites.

**Figure 5 polymers-13-02046-f005:**
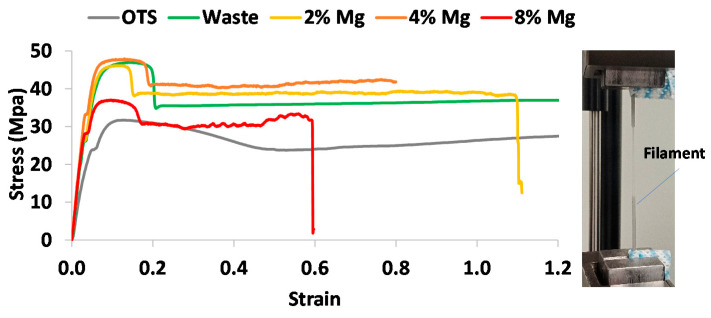
Tensile stress–strain profiles for different prints.

**Figure 6 polymers-13-02046-f006:**
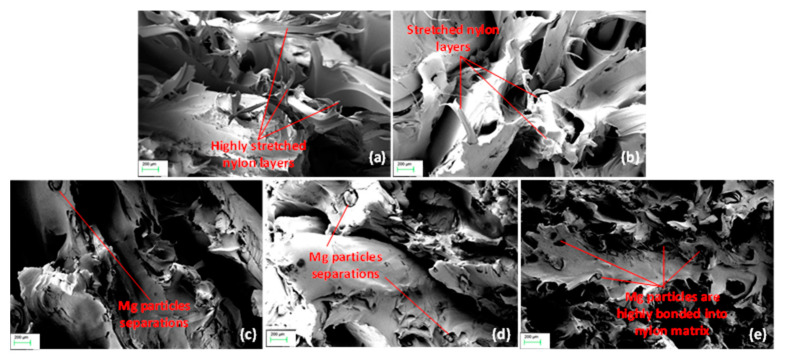
SEM imaging of tensile failure zones for: (**a**) OTS; (**b**) waste; (**c**) 2% Mg; (**d**) 4% Mg; (**e**) 8% Mg dog bone samples.

**Figure 7 polymers-13-02046-f007:**
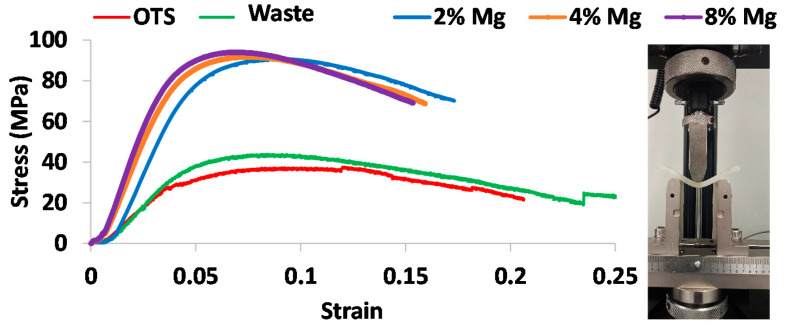
Flexural stress–strain profiles for different prints obtained from three-point bending tests.

**Figure 8 polymers-13-02046-f008:**
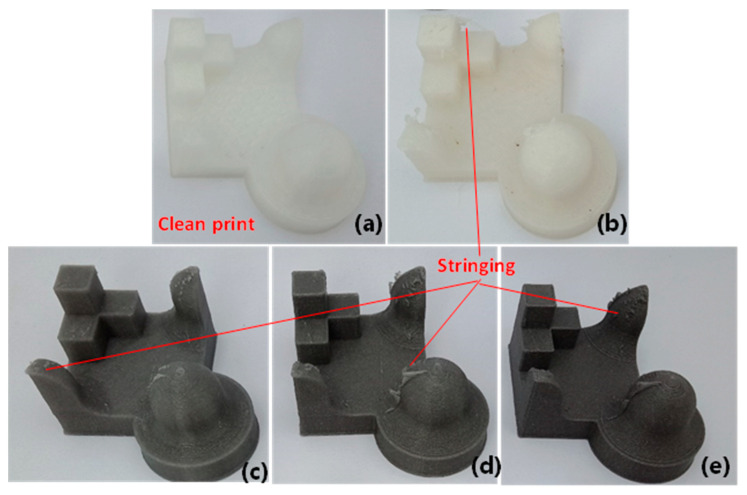
Benchmark print models for: (**a**) OTS; (**b**) waste; (**c**) 2% Mg; (**d**) 4% Mg; (**e**) 8% Mg.

**Figure 9 polymers-13-02046-f009:**
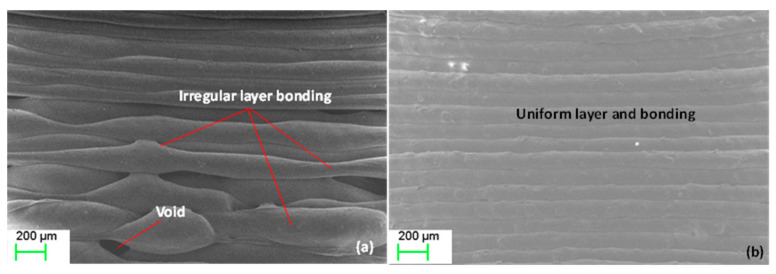
SEM imaging of the interfacing bonding between the layers for: (**a**) waste; (**b**) 4% Mg prints.

**Figure 10 polymers-13-02046-f010:**
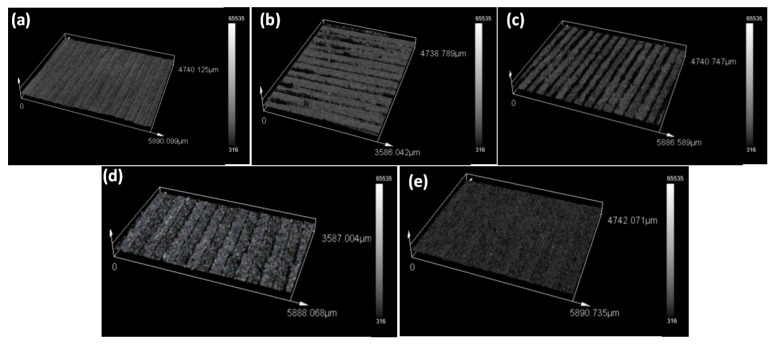
3D surface topography for (**a**) OTS, (**b**) waste, (**c**) 2% Mg, (**d**) 4% Mg, and (**e**) 8% Mg prints.

**Figure 11 polymers-13-02046-f011:**
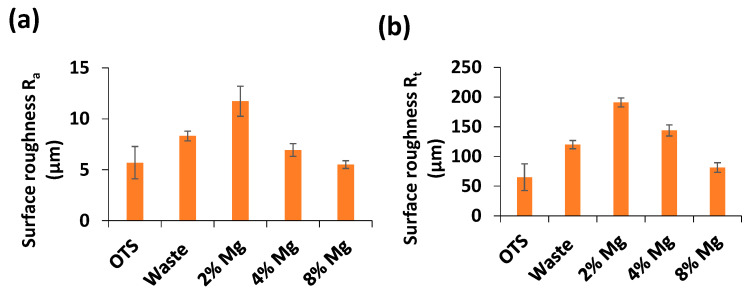
Comparison of surface roughness (**a**) R_a_ and (**b**) R_t_ for different prints.

**Figure 12 polymers-13-02046-f012:**
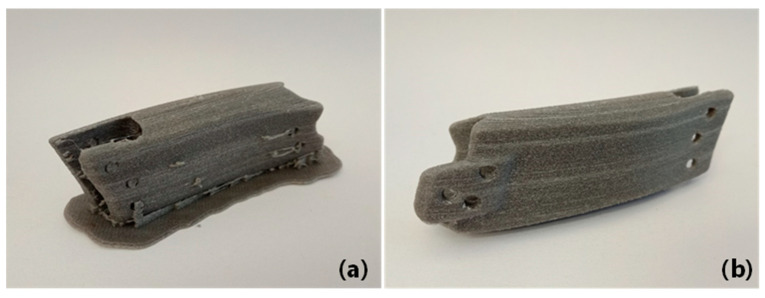
A jaw implant printed with 4% Mg composite filament: (**a**) before finishing; (**b**) after finishing.

**Table 1 polymers-13-02046-t001:** Filament diameter and moisture absorption.

Sample	Type	Diameter * (mm)	Moisture Absorption * (%)
Virgin	Powder	n/a	0.41 ± 0.04
Waste	Powder	n/a	0.42 ± 0.05
Waste	Filament	1.74 ± 0.05	0.38 ± 0.07
OTS	Filament	1.74 ± 0.02	0.95 ± 0.1
2% Mg	Filament	1.74 ± 0.03	0.39 ± 0.20
4% Mg	Filament	1.7 ± 0.02	0.49 ± 0.09
8% Mg	Filament	1.71 ± 0.01	0.31 ± 0.02

* Value ± standard deviation.

**Table 2 polymers-13-02046-t002:** Melt flow index results.

Sample	Type	Melt Flow Index (MFI) * (g/10 min)
Virgin	Powder	52.01 ± 6.49
Waste	Powder	20.38 ± 3.82
Waste	Filament	36.06 ± 2.97
OTS	Filament	5.12 ± 0.99
2% Mg	Filament	22.12 ± 3.23
4% Mg	Filament	19.30 ± 0.92
8% Mg	Filament	15.01 ± 0.64

* MFI value ± standard deviation.

**Table 3 polymers-13-02046-t003:** Summary of DSC test results.

Sample	Type	Crystallisation Temperature *T_c_* (°C)	Melting Temperature *T_m_* (°C)
Virgin	Powder	143.31	178.07
Waste	Powder	147.4	191.7
Waste	Filament	153.25	179.05
2% Mg	Filament	153.84	^a^ 176.56
4% Mg	Filament	153.76	^a^ 176.56
8% Mg	Filament	153.07	^a^ 176.72

^a^*T_m_* at the maximum peak in the second heating cycles for nylon–Mg composite filaments.

**Table 4 polymers-13-02046-t004:** Tensile test results for filament and dog bone samples.

Sample	Type	Yield Strength * (MPa)	Young’s Modulus * (MPa)	UTS * (MPa)	Elongation at Break * (%)
OTS	Filament	23.47 ± 0.2	594.76 ± 27.4	30.96 ± 0.7	^a^ n/a
Waste		24.77 ± 0.5	938.3 ± 47	46.20 ± 1.2	^a^ n/a
2% Mg		27.41 ± 1.5	1108 ± 191.9	46.63 ± 2.9	^a^ n/a
4% Mg		32.23 ± 2.3	1065 ± 54.8	47.04 ± 1.1	^a^ n/a
8% Mg		28.78 ± 1.0	892 ± 119.2	36.69 ± 1.0	^a^ n/a
OTS	Dog bone	27.86 ± 0.6	1240.2 ± 252.9	35.32 ± 1.1	106.36 ± 16.7
Waste		29.36 ± 1.19	1348 ± 178.4	37.38 ± 0.9	21.1 ± 5.7
2% Mg		32.94 ± 0.5	1233 ± 417.8	36.9 ± 2.6	16.25 ± 3.8
4% Mg		36.31 ± 0.04	1480 ± 341	38.81 ± 1.2	14.92 ± 2.2
8% Mg		38.7 ± 0.19	1514 ± 210	39.65 ± 1.3	9.92 ± 1.5

^a^ Tensile filament did not break, hence no data for elongation at break was recorded during filament tests. * Value ± standard deviation.

**Table 5 polymers-13-02046-t005:** Flexural testing results.

Sample	Flexural Strength * (MPa)	Flexural Modulus * (MPa)
OTS	30.91 ± 0.8	202.59 ± 36.9
Waste	34.81 ± 1.7	255.80 ± 10.8
2% Mg	90.81 ± 2.7	1104.83 ± 68.1
4% Mg	92.44 ± 1.9	1239.70 ± 59.5
8% Mg	95.70 ± 3.1	1397.49 ± 133.8

* Value ± standard deviation.

## Data Availability

Not applicable.

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
