# Peer review of "Recycling of Selective Laser Sintering Waste Nylon Powders into Fused Filament Fabrication Parts Reinforced with Mg Particles"

_polymers, 2021, doi:10.3390/polym13132046_

Round 1
Reviewer 1 Report
The authors investigated the feasibility of using waste powder from SLS to create filament for FFF. While the manuscript is generally well executed, there are several issues in the manuscript that should be addressed before further consideration for publication.
- Suggest the authors to use ISO/ASTM terminology when describing the processes.
- For the recycled powder, is there any comparison done with the virgin powder? How much is the difference?
- What causes the change in the viscosity such that the mixed powder can be extruded but not the virgin powder? Any consideration to mix in Mg into the virgin powder for benchmarking?
- Any analysis done to show that the Mg has been mixed in homogenously?
Author Response
Reviewer 1:
The authors investigated the feasibility of using waste powder from SLS to create filament for FFF. While the manuscript is generally well executed, there are several issues in the manuscript that should be addressed before further consideration for publication.
Question: Suggest the authors to use ISO/ASTM terminology when describing the processes.
Reply: The processes – e.g. extrusion of filament, DSC, tensile and flexural tests conducted followed ASTM standards. Those standards are clearly mentioned in corresponding Sections 2.2, 2.3. 2.4, 2.5, as highlighted by blue colour text, of the revised manuscript.
Question: For the recycled powder, is there any comparison done with the virgin powder? How much is the difference?
Reply: Only moisture absorption, melt flow index and DSC were conducted for virgin powder. As shown in Table 1, both virgin and waste nylon don’t show any difference in moisture absorption, but their absorption ability is lower than OTS, giving more shelf life. MFI for virgin power is the highest (52 g/10 min) among all including waste and Mg composites (see Table 2). Therefore, the virgin powders are not extrudable into filaments. This is why, the virgin powders are being used as feedstock in SLS-based additive manufacturing. DSC for waste and composites shows the melting and crystallisation temperatures are higher than the virgin. This is due to the repeated heating and cooling cycles the virgin power underwent during SLS process.
Question: What causes the change in the viscosity such that the mixed powder can be extruded but not the virgin powder? Any consideration to mix in Mg into the virgin powder for benchmarking?
Reply: Chen et al. [1] studied systematical mechanisms of PA-12 nylon aging and its micro-structural evolution during SLS. It was found that the effect of solid-state polycondensation reduces the crystallinity of the powder by ~6% after the reuse of sintered nylon. Pham et al. [2] found that in SLS, the PA12 nylon powder undergoes molecular chain entanglement to form spherulites. The radius of the spherulites is a positive function of time and temperature. The higher the temperature and the longer the powder is exposed to high temperatures, the more likely that the molecular chains become larger, which eventually leads to an increase in molecular weight, a decrease in fluidity, and ultimately the deterioration of the powder’s mechanical and thermal properties. Therefore, the waste powders can be extrudable. With the addition of Mg powders into waste nylon, the viscosity has further increased.
We have added a brief comment on the above in Discussion section of the revised draft.
We have not added Mg into virgin powders. The aim of this study is to salvage waste powder into potential FFF prints and to further reinforce with Mg powders to increase mechanical properties. Virgin nylon SLS powders are very expensive, and more suitable for expensive SLS prints with complex geometries. Therefore, the scope of the study was to mix waste nylon with Mg powders.
- Chen, P.; Tang, M.; Zhu, W.; Yang, L.; Wen, S.; Yan, C.; Ji, Z.; Nan, H.; Shi, Y. Systematical Mechanism of Polyamide-12 Aging and Its Micro-Structural Evolution during Laser Sintering. Polymer Testing 2018, 67, 370–379, doi:10.1016/j.polymertesting.2018.03.035.
- Pham, D.T.; Dotchev, K.D.; Yusoff, W.A.Y. Deterioration of Polyamide Powder Properties in the Laser Sintering Process. Proceedings of the Institution of Mechanical Engineers, Part C: Journal of Mechanical Engineering Science 2008, 222, 2163–2176, doi:10.1243/09544062JMES839
Question: Any analysis done to show that the Mg has been mixed in homogenously?
Reply: Fig. 1 shows photos of Mg-waste mixture. We can see a proportional change in color as we increase Mg % in the mixture, i.e. the color of mixture changes from white to dark grey. Also, SEM image in Fig. 2c exhibits the presence of relatively a uniform distribution of Mg and waste powders within the mixture. These analyses indicate an acceptable homogeneity of mixture.
Reviewer 2 Report
The manuscript of Uddin et al manuscript focuses on the processability of nylon powder wastes from a selective laser sintering (SLS) process. This waste is compared in terms of morphological aspects, thermal and mechanical properties and fluidity, with mixtures containing 2%, 4% and 8% by weight of magnesium powders as well as with off-the-self (OTS) nylon in order to verify its recyclability in filaments for 3D printers intended for biomedical applications.
The research is very interesting and has been described in very clear language. The results, properly and exhaustively discussed, allow an advancement of knowledge on this topic of great attraction both for the enormous quantity of waste powders coming from the SLS technology and for the growing industrial sensitivity towards compliance with appropriate circularity indicators ever more stringent for the global competitiveness of the manufacturing sector.
In light of these considerations, the manuscript is recommended for publication as received.
Author Response
Reviewer 2:
The manuscript of Uddin et al manuscript focuses on the processability of nylon powder wastes from a selective laser sintering (SLS) process. This waste is compared in terms of morphological aspects, thermal and mechanical properties and fluidity, with mixtures containing 2%, 4% and 8% by weight of magnesium powders as well as with off-the-self (OTS) nylon in order to verify its recyclability in filaments for 3D printers intended for biomedical applications.
The research is very interesting and has been described in very clear language. The results, properly and exhaustively discussed, allow an advancement of knowledge on this topic of great attraction both for the enormous quantity of waste powders coming from the SLS technology and for the growing industrial sensitivity towards compliance with appropriate circularity indicators ever more stringent for the global competitiveness of the manufacturing sector.
In light of these considerations, the manuscript is recommended for publication as received.
Reply: The authors thank the reviewer for the comments.
Round 2
Reviewer 1 Report
NIL